# Persistent Organic Pollutants and Fatty Acid Profile in a Typical Cheese from Extensive Farms: First Assessment of Human Exposure by Dietary Intake

**DOI:** 10.3390/ani12243476

**Published:** 2022-12-09

**Authors:** Cristina Giosuè, Fabio D’Agostino, Giuseppe Maniaci, Giuseppe Avellone, Marzia Sciortino, Viviana De Caro, Adriana Bonanno, Marialetizia Ponte, Marco Alabiso, Antonino Di Grigoli

**Affiliations:** 1Institute for Anthropic Impacts and Sustainability in the Marine Environment, National Council of Research (IAS-CNR), Lungomare Cristoforo Colombo 4521, Loc. Addaura, 90149 Palermo, Italy; 2Institute for Anthropic Impacts and Sustainability in the Marine Environment, National Council of Research (IAS-CNR), Via del Mare n. 3, Torretta Granitola, 91021 Trapani, Italy; 3Department of Agricultural, Food and Forest Sciences (SAAF), University of Palermo, Viale Delle Scienze n. 13, 90128 Palermo, Italy; 4Department of Biological, Chemical and Pharmaceutical Sciences and Technologies (STEBICEF), University of Palermo, Via Archirafi 32, 90123 Palermo, Italy

**Keywords:** polychlorinated biphenyls, polycyclic aromatic hydrocarbons, polybrominated diphenyl ethers, autochthonous cow breed, pasture, production season, Caciocavallo Palermitano cheese, chemical composition, fatty acids

## Abstract

**Simple Summary:**

Dairy products have a key role in the human diet due to different healthy traits. At the same time, they can contain different environmental pollutants, representing a risk to human health. The dairy characteristics influencing the risk–benefit ratio are affected principally by animal diet. This paper investigates typical stretched cheese, which is obtained from the milk of the Cinisara cattle breed, mainly raised on pasture integrated with various feed, depending on pasture availability. The present study investigated the fatty acid profile and the persistent organic pollutants content in cheeses made by six typical farms, two of which adopted an organic system. The cheeses were made in winter, spring, and summer. The aim of research was to assess the risk and benefits to human health due to cheese consumption. The results showed a better fatty acids profile of cheeses made in spring, for the presence of some fatty acids deriving from grazing fresh forage; higher contaminants were found in products made in winter, especially in those from non-organic farms. The consumption of different cheeses could represent human health risks, mainly from polychlorinated-biphenyl contents. Further studies should be conducted to identify the pollutants’ pathways and transfer routes due to ingestion.

**Abstract:**

Dairy products represent an important source of beneficial substances for humans. At the same time, they can expose the consumers to environmental contaminants ingested by animals through their diet, influencing their health negatively. This experiment aims to evaluate the risk and benefits related to the consumption of typical stretched cheeses, considering their fatty acid (FA) profile and persistent organic pollutants (POPs) content. Six representative farms, two of them organic, raising Cinisara cattle were selected, considering the typical extensive management systems, based on feeding of natural pasture integrated with concentrate and hay depending on the availability of forage on pastures. A total of 18 cheeses produced in winter, spring and summer with bulk milk of each farm were sampled and analyzed. The chemical composition of cheeses was influenced by farm management, and the FA profile mainly by the season. In particular, cheeses made in spring showed a healthier FA profile with the content of polyunsaturated fatty acids (PUFA), of omega3-PUFA and omega6/omega3 ratio pair to 7.29%, 1.44% and 1.32, respectively, while in winter 5.44%, 0.98% and 2.55, respectively, and in summer 4.77% 0.49% and 3.04, respectively. Due to high levels of feeding integration, cheese made in winter presented unhealthier characteristics compared to the cheeses made in spring and summer, showing high levels of saturated FA (66.2%, 64.2% and 65.5%, respectively), and large contents of polycyclic aromatic hydrocarbons (PAH) (57.07 ng/g fat, 36.25 ng/g fat and 10.22 ng/g fat, respectively) and polychlorinated biphenyls (PCBs) (36.19 ng/g fat, 4.68 ng/g fat and 3.73 ng/g fat, respectively), mainly in those from non-organic farms. Levels of PCBs considered to be hazardous to human health were found in nine samples.

## 1. Introduction

Many works show the risks involved due to the occurrence of persistent organic pollutants (POPs) in the environment, affecting negatively both ecosystems and human health [1,2,3,4,5,6,7,8,9,10,11,12,13,14]. Only a few investigations were carried out on transfer of POPs to cow milk and the subsequent human risk by ingestion of dairy products [1,2,4,6,8,11,13]. Despite the progressive improvement of risk management measures, different pollutants from natural and anthropogenic activities can be released from various sources, where they persist as results of recent or past releases [1,2,3,4,5,6,7,8]. This category includes the persistent organic pollutants (POPs), such as the polycyclic aromatic hydrocarbons (PAH), the polychlorinated biphenyls (PCBs), but also the halogenated flame retardants (HFRs), as the polybrominated diphenyl ethers (PBDEs). When they reach soil, water, and pasture from atmospheric deposition of local emission sources, they can enter the food chain consumed by livestock and, consequently accumulate in food rich in fat (such as meat, fatty fish, milk and dairy products), thanks to their high lipophilic character [1,2,3,4,5,6,7,8,9,10,11,12,13,14]. Therefore, the foodstuff can present complex mixtures of these pollutant compounds, related to the environmental quality and animal exposure, causing bioaccumulation consequently in humans due to metabolism capacities [1,2,3,4,5,6,7,8,9,10,11,12,13,14,15,16,17,18,19,20,21,22,23,24,25,26,27]. Their intake via contaminated products can determine different adverse effects on human health [4,8,9,10,13,14], with relevant factors including the kind of contaminant, exposure and individual conditions, causing different diseases, such as infertility, hormonal and reproductive system disorders, immunological and neurological toxicity, and carcinogenic outcomes [4,8,9,10,13,14]. Milk and dairy products are widely consumed foodstuffs and represent critical pathways for PAHs, PCBs, and PBDEs.

The animals can ingest various environmental matrices, such as pasture, hay, feed, water and soil; the latter represents a minimum of 10% daily dry matter, even up to 30% in the worst conditions in winter, in free-range agricultural animals [7]. The PAHs in cow milk may be found during the pollution event; because they are generally metabolized by ruminants, their transfer to milk occurs at a low level also when the PAHs ingestion is high by feeding [4]. Furthermore, industrial processes can produce PAHs in food (such as smoking, drying and heating), as well as cooking methods (such as roasting, baking, frying and grilling), or by release from packaging materials [4]. Considering the exposure to PCBs and PBDEs, the metabolism processes are more complex than those of PAH. They can be found with different patterns in milk, even over time, as a function of the degradation susceptibility, the absorption efficiency, or the transfer capability of animal species [7,28]. Different studies were carried out on PCBs carry-over rates, showing a variation from 0.2% to 77%, with high levels in cow milk grazing contaminant areas [2,7].

At the same time, the consumption of pasture can improve the quality of animal products, improving the fatty acids profile and transferring nutraceutical compounds in milk and cheese with consequential beneficial effects on human health [28,29]. These molecules are either present in grass, such as docosapentaenoic acid (DPA), or are produced by the animals from their precursors in the grass, such as α-linolenic acid (ALA) for conjugated linoleic acid (CLA), eicosapentaenoic acid (EPA), docosahexaenoic acid (DHA), which are fatty acids present only in animal products. 

Even though natural resources can, therefore, transfer both pollutants and bioactive substances to products of animal origin, there is a lack of studies on the risk and benefits for humans related to their consumption of different foodstuff categories, such as dairy products. These aspects are often separately investigated, while they should be considered together, for a better evaluation of the quality of products especially obtained from animals which are open-raised.

Feeding systems based on pasture are typically adopted by small farms, rearing autochthonous breeds, able to exploit natural resources in areas characterized by different environmental constraints, producing in harsh conditions where the specialized breeds fail to express their genetic potential [30]. The Cinisara is a dual-purpose autochthonous breed of dairy cows with a marked rusticity that allows the breed to exploit the marginal areas in Palermo, Messina, and Trapani provinces (towns in Sicily, Italy). Feeding is based on natural pastures with the integration of hay and concentrates according to the availability of grazed forage and animal productivity. During summer, integration with Opuntia ficus-indica cladodes is recurrent [31]. The milk is used for the production of a typical stretched cheese, called Caciocavallo Palermitano, according to the traditional method [32,33], while the meat is used for fresh consumption and recently also for the production of bresaola and salami [34,35]. Considering the lack of information on the risk and benefits for humans, related to the consumption of cheese, considering both their persistent organic pollutants (POPs) content and the fatty acid (FA) profile, a preliminary study was carried out on this typical stretched cheese, made by six different farms (two of which adopt an organic system) in different seasons, determining both these substances’ contents and then human exposure by dietary intake.

## 2. Materials and Methods

### 2.1. Sampling

A total of 18 bovine cheeses (6 in winter, 6 in spring and 6 in summer), from 6 farms (2 organic farms) located at 550–950 m above sea level (a.s.l.) in the typical production area of Western Sicily (Italy) of the Cinisara cow, were collected in December, August and April (2018–2019). About 500 g of cheese sample, obtained from a 30-day aged cheese shape of about 7–8 kg (made by 70–80 L of bulk milk), was obtained and lyophilized in the laboratory (Italian Regulation n. 229 of 2th October 1986—methods for analysis of dairy products). Different aliquots were used for the analysis of fatty acids and contaminants as described in the following paragraphs. 

During the same day of the sampling, data on the management system were recorded. 

The animals were reared adopting an extensive system based on feeding in natural pastures integrated with hay produced by the fam and commercial concentrate, depending on the pastures’ availability. In particular, the conventional farms used the same concentrate, as well as the organic ones. 

In particular, in winter (November–February) and in summer (July–September), all the farms integrated the animal diet with concentrate and hay or only concentrate, while in spring (March–May), the pasture was the prevalent resource for most farms. The specific farm’s characteristics and formulation of diets administrated indoors are reported in Table 1.

### 2.2. Cheese Characterization

Dry matter (DM), crude protein (CP, N × 6.38), fat, ash and NaCl content were determined according to International Dairy Federation (IDF) standards [36,37,38,39,40] on cheese samples.

Fatty acids (FA) in lyophilised cheese samples (100 mg) were directly methylated as described by Loor et al. [41]. Fatty acid methyl esters (FAME) were recovered in hexane (1.5 mL). An autosampler injected each sample (1 μL) into an HP 6890 gas chromatography system equipped with a flame ionization detector (Agilent Technologies Inc., Santa Clara, CA, USA). The separation and identification of each FA were performed as described by Di Grigoli et al. [42]. The health-promoting index (HPI) was calculated as suggested by Chen et al. [43], following the formula reported below:$$\mathrm{HPI}=\frac{\mathrm{PUFA}\;+\;\mathrm{MUFA}}{C12:0+\left(4\cdot C14:0\right)+C16:0)}$$

The thrombogenic index (TI) was calculated according to Ulbritch and Southgate [44], as below reported: $$\mathrm{TI}=\frac{C14:0+\;C16:0+\;C18:0}{\left(0.5\cdot \Sigma \mathrm{MUFA}\right)+\left(0.5\cdot \Sigma \mathrm{PUFAn}6\right)+\left(3\cdot \Sigma \mathrm{PUFAn}3\right)+\left(\Sigma \mathrm{PUFAn}3/\Sigma \mathrm{PUFAn}6\right)}$$

### 2.3. Contaminants Materials

Methanol, chloroform, hexane and dichloromethane solvents pesticide grade were purchased from VWR (Milano, Italy). SPE Florisil 1 g/6 mL and KOH pellets were from Supelco.

The standard mixes used to calibrate the instrument were: (i) poly-chlorinate biphenyl mix (PCB mix contained PCB28, PCB52, PCB101, PCB81, PCB77, PCB123, PCB114, PCB118, PCB105, PCB153, PCB138, PCB126, PCB128, PCB156, PCB157, PCB167, PCB180, PCB169, PCB170, PCB189, PCB209, each component at 20 μg/g in hexane), (ii) polycyclic aromatics hydrocarbon (PAH—Mix9 contained Naphthalene, Acenaphthylene, Acenaphthene, Fluorene, Phenanthrene, Anthracene, Fluoranthene, Pyrene, Benzo(a)Anthracene, Chrysene, Benzo(b)Fluoranthene, Benzo(k)Fluoranthene, Benzo(a)Pyrene, Indeno(123cd)Pyrene, Dibenzo(ah)Anthracene, Benzo(ghi)Perylene, each component at 10 μg/g in hexane) obtained from Dr. Ehrenstorfer GmbH, (iii) poly-bromurate diphenyl ethers (PBDE mix containing BDE28, BDE47, BDE66, BDE85, BDE99, BDE100, BDE153, BDE154, BDE183, each component at 10 μg/g in acetone), all mixes purchased from Dr Ehrenstorfer GmbH. 

The standard mixtures used as an internal standard or to spike the samples were: (i) PAH deuterated Mix 77 (containing Acenaphthylene D8, Benzo(a)pyrene D12, Pyrene D10), (ii) PAH deuterated Mix 25 (containing Acenaphtene_D10, Chrysende_D12, Perylene_D12, Phenanthrene_D10), (iii) PCB105_C13, BDE47_C13, purchased from Dr Ehrenstorfer GmbH.

### 2.4. Contaminants Extraction

A total of 5 g of lyophilised sample was weighed in a 60 mL glass vial, spiked with a mix of deuterated PAH (Mix 25) and a PCB105_C13 to a final concentration of 1 ng/g and held in freeze overnight. Then, lipids and contaminant compounds were extracted following four main steps:lipids extraction following the Bligh and Dyer method [45];lipids hydrolyzation with alkaline solution 6M (KOH in water) at 80 °C in an oven overnight, and recover the not-hydrolysable contaminants (PAH, PCB, PBDE) by liquid–liquid extraction using dichloromethane (DCM);samples cleaning up using a Florisil SPE (solid phase extract from Supelco) of 1 g/6 mL;solvent evaporation using a multi-vapour (from Buchi coupled with a Buchi Rotavapor) until dry and, in the end, 1 mL of hexane, containing PAH mix deuterated and PCB105C13, both used as internal standard, was added in the test tube. The mixture after stirring was poured into the vials to the instrument analysis.

A total of 16 PAHs, 21 PCBs, and 8 PBDE were quantified, in two different steps (one for PAH and a second for PCBs and PBDEs), using a gas chromatograph coupled with a triple quadrupole mass spectrometer GC-MS/MS with autosampler (GC Trace 1310, MS TSQ8000, and Triplus RSH with a syringe of 10 μL from Thermo Scientific), injecting 5 μL of sample in a large volume injector (LVI), and using a TG-5MS as capillary column [46,47].

#### Quality Assurance and Control

An artificial matrix was built by collecting the same aliquot obtained from all samples and spiked with deuterated PAH_D (MIX25), PCB105_C13, and BDE47C13, with a final concentration of 100 ng/mL to optimize and check the recovery of the extraction and clean-up methods. Three replicates of this artificial matrix were analyzed, and the recovery ranged between 83–110%, 71–92%, and 86–94% for PAH_D, PCB105_C13, and BDE47C13, respectively. Furthermore, to monitor all analytical processes, each sample was spiked with the same mix of contaminants deuterated or PCB105_C13 marked above mentioned, with a final concentration of 100 ng/mL, and the recovery per each analysis was calculated inside the range 72–108%, and 65–93%, respectively.

### 2.5. Estimation of Potential Human Health Risks

The POPs concentration detected in cheese was compared with the maximum admissible concentration (MAC) limits reported by the European Regulations. The Regulation 835/2011 [48], amending Regulation (EC) 1881/2006 [49] for PAH maximum levels in foodstuffs, did not provide specific limits for PAH in cheese, but only for infant and follow-on formulae including infant milk and follow-on milk (MAC: 1 μg/kg for the benzo(a)pyrene or the sum of benzo(a)pyrene, benzo(a)anthracene, benzo(b)fluoranthene and chrysene). Differently, the Commission Regulation (EU) No 1259/2011 (amending Regulation EC No 1881/2006 as regards maximum levels for dioxins, dioxin-like PCBs and non-dioxin-like PCBs in foodstuffs) [50] provided limits for raw milk and dairy products, including butter fat. For these categories, the following limitations were imposed: -sum of dioxins (WHO-PCDD/F-TEQ) 2.5 pg/g fat; -sum of dioxins and dioxin-like PCBs (WHO-PCDD/F-PCB-TEQ) 5.5 pg/g fat; -sum of PCB28, PCB52, PCB101, PCB138, PCB153 and PCB180 (ICES–6) 40 ng/g fat.

The health risk for the resident population due to cheese consumption was assessed based on a standard protocol [51], considering the contaminant levels detected in cheese and the consumption rate reported by INRAN [52].

In specific, the evaluation was run using the combined toxicity effect quotient (TEQ) of each type of organic pollutant family, as shown in Equations (1) to (3), where TEF(i) are indicated by the World Health Organization (WHO) or US EPA. They take into account the relative toxicity of the same pollutant family. Hence, the toxicity of (i) PAHs was calculated in terms of benzo(a)pyrene_TEQ (BaP), namely BaP_TEQ [52]; (ii) dioxin-like polychlorinated biphenyl (dl-PCB) was referred to tetrachloro-dibenzo-p-dioxin (TCDD), namely dl-PCB_TEQ, (iii) while non-dioxin-like polychlorinated biphenyl (ndl-PCB) was calculated easily as a summary of the concentration of each congener.

The equations used were:BaPy_TEQ = Σ C_PAH_i × TEF_BaPy_i, (1)
dl-PCB_TEQ = Σ C_dl-PCB_i × TEF_TCDD_i, (2)
Σ ndl-PCB = Σ (PCB28, PCB52, PCB101, PCB138, PCB153) (3)
where C is the pollutant concentration in cheese (mg/kg), TEF(i) is the toxicity effect factor for the i-th compound (equivalent to BaP toxicity or Tetrachlorodibenzodioxin TCDD), C_PAH_i, C_dl-PCB_i are the concentrations of each “i-th” congener’s cheese occurrence.

The cancer risk (CR), hazard quotient (HQ), and hazard index (HI) were calculated following Equations (4) to (7) [47,51]:ADD = C_food × IR × EF × ED/AT (4)
CR = ADD × CSFo (5)
HQ_i = ADD/(RfDo_i) (6)
HI = ΣHQ_i (7)
where ADD is the average daily dose (mg/kg/day), C food is pollutant concentration in food (mg/g), IR is the ingestion rate per kg body weight, and per days (0.88 g/kg BW per day) [52], EF is the frequency of exposure (days/year), ED is the exposure duration (years), AT is the average lifespan that for CR risk is 2550 days (equal to 70 years × 365 days) while for HQ is 10950 days (equal to 30 years × 365 days), CSFo is oral cancer slope factor (mg/kg per day), RfDo is oral reference dose (mg/kg per day), and gastrointestinal absorption ABS_GI (Table 2).

Based on Equations (5)–(7), the risk level according to the US EPA classification [57,58] was evaluated. When the cancer risk (CR) value (Equation (5)) is less than 1 × 10^−6^, between 1 × 10^−6^ to 1 × 10^−4^, and greater than 1 × 10^−4^, the risk level is negligible, potential/possible, and high, respectively.

The evaluation of non-cancer risk level depends on the calculated values of HQ and HI, Equations (6) and (7), and if they are lower or greater than 1, reflecting no risk and possible risk, respectively.

### 2.6. Statistical Analysis

The data were statistically analyzed by the SAS 9.2 software [59], using a generalized linear model (GLM) that included the effects of production season (S, with three levels: winter, spring, and summer) and farm (F, with six levels: A, B, C, D, E, F). The interaction S × F was removed from the model since it was not significant. Results are reported as LSM and differences between means were tested by Student’s *t*-test. Statistical significance was attributed to *p* values < 0.05.

To evaluate the specific contribution of each variable in explaining the differences between cheeses due to the different production season, a principal component analysis (PCA) was carried out with the PRINCOMP SAS procedure, using variables related to chemical and FA composition. The variables included in the analysis were standardized by multiplying them by the inverse of the standard deviation (1/SD) and identified by gradual selection with the STEPDISC SAS procedure. The selection of the main components was carried out according to the Kaiser method, keeping those with eigen values higher than 1.00.

## 3. Result and discussions

### 3.1. Caciocavallo Cheese Characteristic and Fatty Acid Profile

The chemical composition of Caciocavallo cheese is reported in Table 3. Concisely, the range values determined on eighteen cheese samples were: DM 59.7–68.1%; protein 44.6–48.4%; fat 39.9–45.3%; ash 6.2–8.8%; NaCl 2.31–5.64%, respectively. These results showed similar values to those observed by other authors [31,32,42].

Statistical analysis showed significant differences between the cheeses produced by farms for ash and NaCl. In particular, the NaCl content (% DM) was higher (*p* < 0.001) in cheeses made by farms F (5.64) > B_org (5.15) > (4.76) than by farms C (2.70) > D (2.56) > A_org (2.81). As expected, a similar trend was registered for ash (% DM) showing higher values *p* < 0.01), in products made in farms B_org (9.40) > (9.10) > (8.93) than C (7.13) > D (7.13) > A_org (6.43). Probably, although all the farms carried out the salting in saturated brine, the different times adopted influenced the NaCl absorptions by the cheeses. However, this result can be related to DM content, even if the latter was not significant. In fact, between individual farms, NaCl was higher in cheeses with a higher DM level, as also observed in previous studies [42,60], in contrast to what was found between the various seasons.

Table 4 shows the cheeses’ FA composition in relation to production season and farm. The cheeses were high in palmitic (C16:0) and oleic acids (OA, C18:1c9) and, secondarily, by myristic (C14:0) and stearic acids (C18:0), complying with other studies carried out on the same product [31,32]. FAs contents and profile on cheese samples were influenced mainly by season rather than farm.

Statistically, the farm influenced only the content in C15:0 (*p* < 0.01), in C18:2 n6 (linoleic acid, LA) (*p* < 0.05), in n6 PUFA (*p* < 0.05) and n6/n3 (*p* < 0.05). In particular, LA content was higher (*p* < 0.05) in farm D (2.14%) than in farms E (1.52%), C (1.49%) and B_org (1.36), and in farm F (1.93%) compared to farm B_org (1.36%). This determined an increase (*p* < 0.05) of n6 FAs in farms D (2.58%) and F (2.30%) compared to farms C (1.93%), E (1.89%) and B_org (1.84%). Although not significant, the higher n6 content in D and the lower n3 content resulted in a higher n6/n3 ratio (*p* < 0.05) in D (3.44) compared to A_org (1.77) and C (1.62). 

Considering the season, most of the short and medium chain FA were significantly higher in cheeses made in winter and spring. In particular, butyric acid (C6: 0), propionic acid (C8: 0), capric acid (C10: 0) and lauric acid (C12: 0) were higher in spring (2.33%, 1.46%, 3.07% and 3.37% respectively) and in winter (2.28%, 1.39%, 2.80% and 3.05%, respectively) than in summer (1.19%, 1.08%, 2.15% and 2.37, respectively). As observed by Di Grigoli et al. [42] in the same area, the low levels of these FAs in summer milk are attributable to the reduction of their de novo synthesis in the udder, related to the quantitative and qualitative deterioration of pastures, together with unbalanced feeding integrations provided to cows. Moreover, the inadequate diets should lead to a mobilisation of body fat reserves, resulting in the higher content of OA observed in cheeses produced in winter and summer (*p* < 0.01), as also found by Chilliard et al. [61].

In the Mediterranean environment, spring is typically characterised by high availability of forage in the pastures, reducing feeding integrations, especially when the farms raise autochthonous breeds. In spring, the increased intake of cows grazing fresh forage led to a significant increase, compared to other seasons, in C18:1t11 (trans vaccenic acid, TVA), other C18:1, C18:2c9t11 (rumenic acid, RA), other isomers of conjugated linoleic acid (CLA) and C18:3n3 (α-linolenic acid, ALA), as observed by other authors [31,42,62]. 

The RA and ALA are very important for their positive effects on human health. The consumption of cheeses naturally enriched with RA-induced positive biochemical changes in atherosclerotic markers [63], as well as the products containing CLA, determine a reduction of the endocannabinoid anandamide and LDL (cholesterol level) in plasma concentrations of hypercholesterolemic subjects [64]. The ALA is a fatty acid present in high quantities in fresh forage and transferred in milk and cheese. However, ALA is in part biohydrogenated in the rumen, determining a TVA increase, that in the mammary gland is oxidated by the delta-9 desaturase to RA [62,65,66,67]. 

Furthermore, the greater amount of fresh grass, that was probably ingested by the animals in spring, has resulted in a higher content of C22:5 (docosapentaenoic acid, DPA) in cheeses (*p* < 0.05), an important long-chain n3 FA which plays a role in reducing blood sugar [68] and in the prevention and treatment of cardiovascular diseases [69].

Thus, the greater availability of fresh forage in spring has also increased the content of polyunsaturated fatty acids (PUFA) and n3 PUFA in cheeses (*p* < 0.01). On the other hand, the higher LA content of winter and summer cheeses led to an increase in the n6 PUFAs levels in these products. Consequently, the n6/n3 ratio was more favourable in cheeses made in spring than in the other seasons (*p* < 0.01), as observed by Altomonte et al. [70], and always lower than 5 (limit indicated by the FAO/WHO) [71]. LA and ALA being the precursors of the n6 and n3 series of FAs, respectively, represent the simplest members of each PUFA family and are called essential fatty acids because the body is unable to synthesize them. Numerous studies report the related health benefits of n3 PUFAs and their effects on cardiovascular disease, diabetes, cancer, Alzheimer’s disease, and immune function [72].

The PCA plot is shown in Figure 1, reporting each selected variable on the main components with a vector of length proportional to its contribution.

The first two principal components accounted for 67.77% of the total variance, allowing for the discrimination of the cheeses for production season. At the same time, they determined a non-linear separation of the farms within the seasonal groupings. The first main component (39.06% of the total variance), with the contributions of DM, Protein, Ash and n6/n3, had a minor impact in discriminating cheeses based on the production season. Instead, the second principal component (28.71% of the total variance) was more responsible for the separation of cheeses based on the production season, mainly due to the contribution of OA, C16:0, TVA and, to a lesser extent, C18:0, C14:0, RA and C12:0.

### 3.2. PAH and PCB Congeners, Abundance and Possible Source 

The concentration of 16 PAH congeners (Appendix A), 20 PCB congeners (Appendix A), and 9 PBDE congeners were investigated in eighteen cheese samples. PBDE results were for all samples per each congener below the limit of detection, so the data was not reported in the tables.

Referring to each congener of PAH (Appendix A), certain contaminations of naphthalene (range: 0.13–30.49 ng/g) > phenanthrene (range: 0.01–13.01 ng/g) > fluorene (range: 0.11–6.65 ng/g) > acenaphthene (range: 0.01–3.46 ng/g) > pyrene (range: 0.01–1.36 ng/g) > fluoranthene (range: 0.01–1.30 ng/g) and > anthracene (range: 0.01–0.66 ng/g) were found in cheese samples, and the highest concentrations were determined in winter. In general, the low molecular-weight ones (PAH_LMW_) were greater than the high molecular-weight ones (PAH_HMW_), most of them < LOD; indeed, the ratio between PAH_LMW_ and PAH_HMW_ was always greater than 1. The PAHs are generated from pyrolysis and due to incomplete combustion of organic matter [22,73] and humans and animals can be exposed to these contaminants through different routes [74]. The principal ones are food, air, and, more generally, several environmental matrices. Humans are also exposed due to smoking. The PAHs are classified as genotoxic and possibly/probably carcinogenic to humans, and the benzo(a)pyrene (BaP) is the most studied, being classified as a human carcinogenic in Group 1 [75]. According to the EU Scientific Committee on Food, the BaP and ∑4PAHs (benzo(a)pyrene + benzo(a)anthracene + chrysene + benzo(b)fluoranthene) can be used as a marker for carcinogenic PAHs in food [48,49], providing maximum levels only for certain foodstuff categories characterized by high PAH levels, for example, the smoked ones. Indeed, the food can be contaminated directly: by the PAH occurrence in air, soil and water, by industrial emission, and by home food preparation (e.g., heating, drying, smoking, grilling and roasting processes) [74]. The animals can be exposed to PAHs mainly by the inhalation of particulate matter, intake of dietary food, and contact with any other materials contaminated by PAHs (such as soil, which could be more contaminated than fodder and daily ingested in a percentage from 1 to 30 % by grazing the pasture) [7]. These pollutants are largely excreted in urine or faeces in a hydroxylated form, due to the metabolization process, but are also absorbed in the body [76,77]. After the animal exposure to PAHs, due to their lipophilicity, these can be accumulated in animal adipose tissue and milk relative to the complex mobilisation process of energetic animal resources during the lactation and the gestation periods [76,78] and, consequently, transferred in cheeses. In the present study, as described above, the low molecular-weight PAHs (PAH_LMW_) in cheeses were greater than the high molecular-weight ones (PAH_HMW_) for different reasons. The forage usually presents PAH_LMW_ > PAH_HMW_ [77,79,80], and it was the feeding basis for the animals involved in the trial. Indeed, the PAH_LMW_ tend to adhere to the intracuticular wax of the plants differently than the PAH_HMW_, which are found in the epicuticular wax and so more exposed to photodegradation and washing [78,81]. Moreover, the PAH_LMW_ can also be transported over long distances [78,82] in the gas form [78], differently to PAH_HMW,_ that falls out near the emission site [83], mainly in condensate form. Consequently, the PAH_HMW_ are generally observed in pasture forage and soil only when the contaminating source is powerful and consistent and the plants are very close to it [79,80]. This study did not investigate the contaminants in pasture and feed, but other authors found an abundant presence of specific PAH_LMW_ in grasses, such as naphthalene, phenanthrene, fluoranthene, and pyrene [77,79,80], the same congeners found in the investigated cheeses. Few PAH_LMW_ detected in the present cheeses are transferred from feed and soil to milk at low levels (transfer rate from 0.5 to 8%) because most PAHs are probably bio-transformed and excreted by urines [77,78,84,85]. For dairy products, maximum levels of PAHs are not imposed, therefore the levels of PAHs found in this study were compared with those detected in previous investigations in unsmoked cheeses, showing a similar trend in the contamination of naphthalene > phenanthrene > fluorene > acenaphthene > pyrene > fluoranthene > and > anthracene, as well as the occurrence of heavy PAHs, most of them lower than LOD [86,87,88]. 

Polybrominated diphenyl ethers (PBDEs), also investigated in this study, are organic chemicals used as flame retardants in numerous consumer products (for example, home electronics, textiles and items containing polyurethane foam, and so on). Humans can be exposed to PBDEs by different sources, inhalation of house dust, absorption by textiles, and diet. PBDE, due to their lipophilicity, could be bioaccumulated from the environment to different foodstuffs [12,13,89]. In the present study, the cheeses showed PBDEs concentrations lower than the detection limit (0.02 ng/g for each congener), differently from those observed in other investigations carried out in Italy [89] and worldwide [90], reporting high concentration of PBDEs in dairy products.

Polychlorobiphenyls were other persistent organic pollutants investigated in the cheese samples. Two classes represent them, non-dioxins such as PCBs (ndl-PCB) and dioxin-like PCBs (dl-PCB), which may be distributed over hundreds of kilometres from any sources of emissions [78]. In some cases, no differences in contamination were found between fresh forage collected near rural areas and those sampled in industrial sites [91]. Similar to those observed for PAHs, the gas deposit concerns the most-volatile compounds, namely the least-chlorinated PCBs, while the least-volatile compounds are found mainly in the form of particulate deposit [78], transferring to animal milk by ingestion of contaminated feed and soil. The PCBs are persistent and could be accumulated in livestock products, differently to PAHs that are largely metabolised [2,78]. In animal products, these contaminants can achieve different pollution levels, and dairy products represent a considerable portion of total dietary exposure, considering the human diet composition [3]. The cheeses sampled in this study showed a presence of dl- PCBs 114 + 118 (range: 0.01–5.41 ng/g; Appendix A), being that PCB 118 is one of the principal congeners found in cow milk [92,93,94], a raw product used to make Caciocavallo Palermitano cheese. A prevalence of ndl-PCB congeners (1.5 × 10^−1^–2.34 × 10^1^ ng/g) on all PCBs, calculated as a sum of PCB (0.23–24.97 ng/g), was found. Most of the dl-PCBs were lower than the detection limits, and were considered equal to ½ LOD in the human health risk assessment. In particular, PCB 52 (0.01–29.37 ng/g) and PCB 28 (0.20–5.22 ng/g) were found in high amounts in cheeses made in winter (Appendix A), followed by PCBs 138 > 153 > 180 (Appendix A), with a different prevalence of congeners, with respect to those usually observed in cow milk, represented by ndl-PCB 153, 138 and 180 [92,93,94]. However, the prevalence of specific PCB congeners in milk and dairy products is observed [94] and related to various factors. Different studies showed carry-over rates higher than 80% for dl-PCB in milk, and from 5 to 40% for PCB indicators (PCB28, PCB52, PCB101, PCB138, PCB153 and PCB180) [2,7,78]. The PCBs profile in milk is different according to occurrence in the environment, due to levels of pollution [95], but also due to the physiological animal states, in a close relationship between contaminant–animal, including the rich ruminants microflora [3,7,96]. PCB 81 and especially PCB 77 seem to show poor transfer to milk, probably due to metabolism or poor absorption [3,7], and in the present investigation, they were in low concentrations in cheese (0.05 and 0.01 ng/g, respectively). EFSA [11] reported that keeping the daily intake unchanged, the calculated transfer rate of PCBs to milk increases with the time exposure and is highest in steady-state conditions. Anyway, the PCBs content in milk is significantly affected not only by the specific carry-over rate of each congener but also by the lactation stages and, in general, by animal conditions related to the energy balance and, consequently, the fat mobilisation [94]. In general, PCBs 28 and 52 are slowly concentrated in corporal deposits and are low-carried in milk. Their presence tends to increase during lactation, probably for the decrease of the more abundant other congeners that are high-carried and also excreted more rapidly [94]. Moreover, malignant breast lesions can influence the prevalence of specific PCB congeners in milk, as observed for PCB 28 and 52, often associated with these pathologies [96,97]. 

The concentration of summary of 16 congeners PAH and Benzo(a)pyrene_TEQ (BaPy_TEQ), 20 congeners of PCB including the PCBndl, PCB_TEQ (referred to as tetrachloride-dibenzo-para-dioxin—TCDD) were detected in the cheeses (Appendix A).

The ΣPAHs ranged between 0.47 and 27.72 ng/g wet weight (ww), showing lower concentration than those found in unsmoked Caciocavallo (36.70–248.59 ng/g ww) produced in Campania (ITALY) [98]. The low occurrence of the most toxic PAHs determined a BaP_TEQ ranging between 8.01 × 10^−3^ and 3.50 × 10^−2^ ng/g; values were also overestimated, considering that the PAHs concentrations lower than the LOD were considered ½ LOD. The BaPy_TEQ was the parameter used for the human health risk assessment. 

The ΣPCB, Σndl-PCB and PCB_TEQ in cheese ranged between 0.23–24.97 ng/g, 1.5 × 10^−1^–2.34 × 10^1^ ng/g, and 3.32 × 10^−4^–6.79 × 10^−4^ ng/g, respectively. The PCB_TEQ was also used for the human health risk assessment. 

A better comparison of contaminant concentration among cheese samples and among farms and seasons was based on normalised values at fat grams contained in cheese (Appendix A), reported as ΣPAH*, ΣPCB*, Σndl-PCB* and PCB-TEQ*. 

The ΣPAH* ranged between 1.76–105.81 ng/g_fat, and the sampled cheeses most contaminated by PAH were Cheese_63, followed by Cheese_64, both made in winter in two different non-organic farms (C and D, respectively). Indeed, the more contaminated cheese by ΣPAH* was found in farms C > D > F > E > B_org > A_org and produced in winter > spring > summer. These trends were shown in the box plots (Figure 2 and Figure 3) and as results of statistical analysis in Table 5. These results are probably due to the influence of the farm management system (organic versus conventional) and the feeding quality. In particular, farms A_org and B_org were organic and integrated the pasture with hay and concentrate in all seasons, unlike the others adopting this management feeding only in winter. Moreover, among the non-organic farms, only farms D and F supplied the concentrate in the spring season, differently to what was observed in summer, where all non-organic farms gave concentrate to the animals. Due to these results, it can be supposed that the animals of organic farms were probably reared in better environmental conditions and fed in all seasons with concentrate and hay less polluted than those used in non-organic farms. These considerations seem to be supported by the different concentration of pollutants found in cheeses in winter > spring > summer, which is in line with the levels of integration adopted especially for non-organic farms, generally higher in winter than in the other seasons. 

These considerations could also explain the similar results obtained for ∑PCB* detected on products made in farms F > E > C > D> A_org > B_org (Figure 2; Table 5), in highest amounts in winter, followed by spring and then summer (Figure 3; Table 5). 

The ΣPCB* and Σndl-PCB* ranged between 0.85–98.60 and 0.53–91.81 ng/g_fat, respectively, showing a higher prevalence of Σndl-PCB* on all ΣPCB*. Σndl-PCB* concentrations were 58.30 and 91.81 ng/g_fat on Cheese_5 and Cheese_6, made in winter by the farms E and F, respectively. These values exceeded the maximum level of 40 ng/g_fat provided for Σndl-PCB* by Commission Regulation (EU) No 1259/2011 [50].

Moreover, the range of PCB_TEQ* in these cheeses was higher than the sum of PCDD/Fs and dl-PCBs TEQ for “raw milk and dairy products, including butter fat” (lower bound mean: 0.73; upper bound mean: 0.88 pg/g fat) reported from EFSA [11], as result of a European monitoring plan. These results are probably overestimated because most of the dl-PCB determinations were lower than LOD, and the TEQ calculation for the risk assessment was imposed equal to ½ LOD. The regulation EU No 1259/2011 [50] also provided a limit for the sum of dioxins and dioxin-like PCBs (WHO-PCDD/F-PCB-TEQ*) equal to 5.5 pg/g_fat and, in this study, the PCB-TEQ* ranged between 1.30–2.39 pg/g_fat, representing the 24–43% of the maximum admissible concentration. In this sense, more investigations in cheese samples could be necessary to detect also the PCDD/Fs for a more appropriate comparison to the limits imposed by regulations. 

### 3.3. Potential Human Health Risk Assessment for Cheese Ingestion

The potential human health risk assessment, calculated following Equations (5) and (7), based on BaP_TEQ, ∑ndl-PCB and PCB_TEQ contaminant concentrations detected in cheese, and referring to the INRAN [52] database for the individual cheese intake by the Italian population across one year and during a lifetime, was reported in Table 6.

The CR values for BaP_TEQ (CR_PAH_), ∑ndl-PCB (CR_ndl-PCB_), and PCB_TEQ (CR_dl-PCB_TEQ_), ranged between 6.95 × 10^−9^–3.03 × 10^−8^, 1.88 × 10^−5^–3.84 × 10^−5^, and 1.13 × 10^−7^–1.76 × 10^−5^, respectively, showed no-risk (<1 × 10^−6^) for BaP_TEQ (PAH), and probably risk (>1 × 10^−6^ and <1 × 10^−4^) for both PCB_TEQ and ∑ndl-PCB.

The CR_PAH_ showed values two-orders of magnitude less than the threshold of 1E-6, evidencing no potential risk by the ingestion of the sample cheeses.

CR_∑ndl-PCB_ for the cheese samples named Cheese_9, Cheese_3, Cheese_4, Cheese_5, and Cheese_6, showed values greater than 1E-6 (1.31 × 10^−6^, 3.96 × 10^−6^, 3.39 × 10^−6^, 1.25 × 10^−5^, 1.76 × 10^−5^, respectively). Cheese_9 was made in the spring season by farm C, while the others in the winter season by farms C, D, E, and F, respectively. 

The CR_dl-PCB_TEQ_ calculated for all eighteen samples exceeded an order of magnitude with the threshold of 1 × 10^−6^ and ranged between 1.88 × 10^−5^ and 3.84 × 10^−5^. These results are overestimated and should be considered not worrying because most of the dl-PCB determinations were lower than the detection limit and were considered equal to ½ LOD in the human health risk assessment. 

Otherwise, if the CR had been calculated only for consumption of cheeses with PCBs concentration greater than the detection limits, the CR would have ranged between 6.65 × 10^−7^–1.88 × 10^−5^, involving only nine cheeses: Cheese_5 (1.88 × 10^−5^) > Cheese_13 (1.02 × 10^−5^) > Cheese_6 (8.82 × 10^−6^) > Cheese_8 (8.51 × 10^−6^) > Cheese_3 (6.76 × 10^−6^) > Cheese_7 (5.13 × 10^−6^) > Cheese_14 (4.04 × 10^−6^) > Cheese_4 (3.50 × 10^−6^) > Cheese_12 (1.21 × 10^−6^) (data not reported in Table 6 and calculated only for the discussions). 

The human health assessment for a “non-cancer risk” used as indicators the hazard quotient (HQ) and hazard index (HI), which were calculated following Equations (6) and (7). The HQs calculated for B(a)Py_TEQ (HQ_B(a)Py_TEQ_), ∑ndl-PCB (HQ_ndl-PCB_), and PCB_TEQ (HQ_PCB_TEQ_) in all sample cheeses, showed values less than 1 for all samples except for the Cheese_6, in which HQ_ndl-PCB_ was slightly greater than 1 (1.03). These values evidence a non-cancer risk due to the ingestion of this kind of cheese.

Otherwise, the HI values (sum of HQ) evidence possible acute stress (HI > 1) only for the samples named Cheese_5 and Cheese_6 (1.58, 1.66, respectively) due to the contribution of each pollutant, mainly PCB. Indeed, these cheeses (5 and 6) were made in winter in farms E and F, respectively, registering a larger amount of PCBs (as the sum of ∑ndl-PCB and dl-PCB_TEQ) than the others. 

The results on risk assessment suggest that a new and more sensible method should be developed to determine pollutants concentration in cheeses, in particular PCB, often below the LOD when detected by conventional ones. 

## 4. Conclusions

Caciocavallo Palermitano cheeses showed differences in chemical composition as a function of the farms, particularly in salt and ash, probably due to different salting times, despite having used the same making method. Moreover, the products presented significant variations in fatty acids content, with the best profile for human consumption in cheeses made by organic farms and in spring, showing higher content in TVA, ALA, RA, and in other isomers of conjugated linoleic acid (CLA), n3 PUFA, and total PUFAs, lower ratio n6/n3. The first two components of the PCA analysis, considering the chemical composition and the fatty acids profile, allowed for the discrimination of the cheeses for production season. Compared to organic farms, the conventional farms produced cheeses more contaminated by PAH_LMW_, and ndl-PCBs, mainly in winter > spring > summer. In particular, two cheese samples exceeded the limits admissible for non-dioxin-like PCBs. Cancer risk evaluation due to PAHs revealed no risk for human health, while probable risk due for both ∑ndl-PCB and PCB_TEQ, mainly for ingestion of cheeses made in winter in non-organic farms, was assessed. The human health for non-cancer risk registered a slightly alerting value only in one cheese sample for ndl-PCB content, and in two cheese samples made in winter in non-organic farms due to the sum of several contaminants. These results of the human health assessment are probably overestimated and should be considered not worrying because most of the dl-PCB determinations were lower than the detection limit and were imposed equal to ½ LOD in the human health risk assessment. However, considering only the values greater than the LOD, nine samples on eighteen seem to be cancerogenic and toxic due to PCB contamination. The results suggest that a new and more sensible method should be developed to determine pollutants concentration in cheeses, particularly for PCB. Comprehensive studies are required to better explain the transfer rate of the POPs to cheese, taking into account other environmental matrices, such as soil and feed, and a more extensive cheese sampling.

## Figures and Tables

**Figure 1 animals-12-03476-f001:**
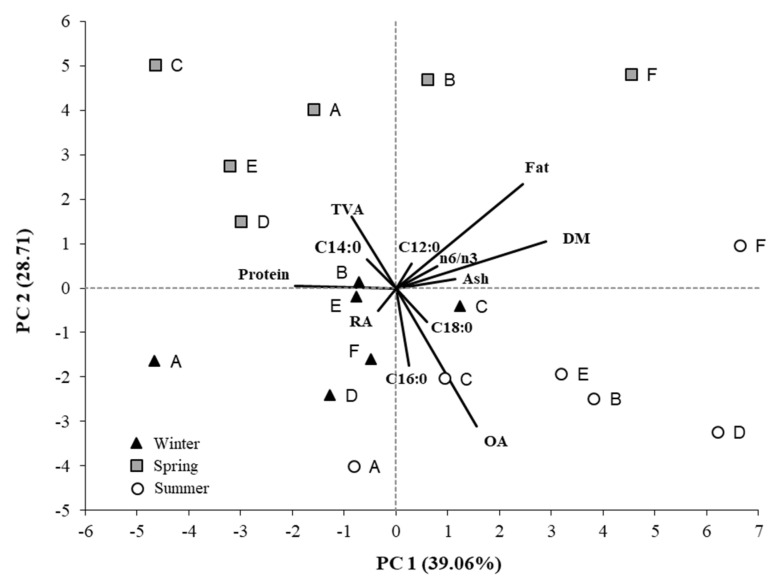
PCA analysis based on chemical traits and FA composition of cheese for production season. The length of each vector is proportional to its contribution to the main components. Abbreviations: DM = dry matter; OA = oleic acid; TVA = trans-vaccenic acid; A (A_org), B (B_org), C, D, E and F = farms.

**Figure 2 animals-12-03476-f002:**
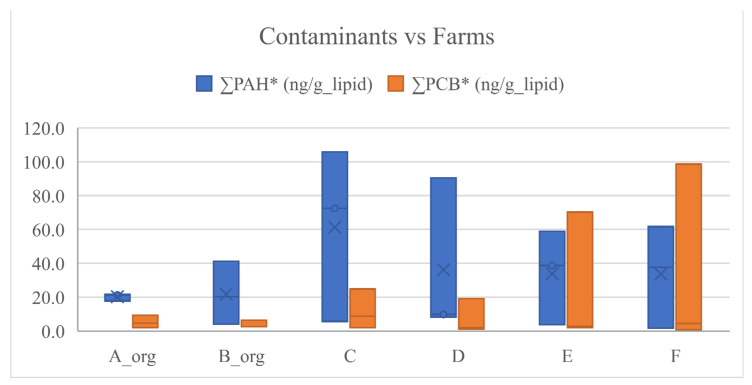
Box plot of contaminants in cheese samples versus farms. The * symbols refer to the values normalized per gram lipid amount.

**Figure 3 animals-12-03476-f003:**
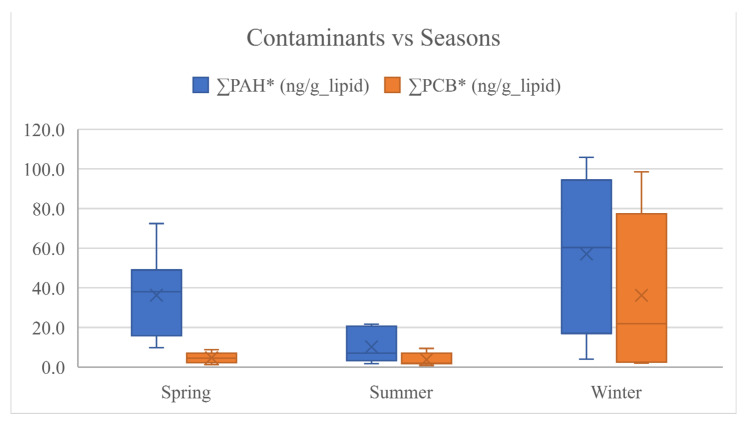
Box plot of contaminants in cheese samples versus seasons. The * symbols refer to the values normalized per gram lipid amount.

**Table 1 animals-12-03476-t001:** Farm characteristics and diet formulation; in square bracket unit measurements.

Farm Characteristics		A_org	B_org	C	D	E	F
Altitude, m a.s.l.		800	550	820	550	950	700
Organic production		yes	yes	no	no	no	no
Available grazing areas, ha		83	170	120	50	150	100
Grazing cattle, number		115	80	120	55	52	120
Indoor diet ingredients							
Season ^1^	Integration	(kg/animal∙day)					
Winter	concentrate	2.0	4.0	4.0	4.0	3.0	4.5
	hay	4.0	2.0	4.0	3.0	4.0	3.0
Spring	concentrate	2.0	3.5	0.0	3.0	0.0	4.0
	hay	5.0	2.5	0.0	0.0	0.0	0.0
Summer	concentrate	3.0	2.0	3.0	4.0	3.0	4.5
	hay	5.0	3.0	0.0	0.0	0.0	0.0

^1^ Winter: November–February; spring: March–May; summer: July–September.

**Table 2 animals-12-03476-t002:** Value of CSFo, RfDo, and ABS_GI used for potential human health risk.

Toxicants	CSFo ^(1)^	RfDo ^(1)^	ABSGI ^(2)^
	(mg/kg per Day)	(mg/kg per Day)	Unitless
Benzo(a)Pyrene	2.30	2.00 × 10^5^	0.89
dl-PCB_TEQ (TCDD) *	1.50 × 10^5^	7.00 × 10^−10^	1.00
ndl-PCB **	2.00	2.00 × 10^5^	1.00

^(1)^ [53,54,55] ^(2)^ [56]. * dioxin-like polychlorinated biphenyl (dl-PCB) was referred to tetrachlo-ro-dibenzo-p-dioxin (TCDD). ** not dioxin-like polychlorinated biphenyl (dl-PCB).

**Table 3 animals-12-03476-t003:** Effect of production season on Caciocavallo cheese characteristics.

	Season (S)			Farm (Fa)						RMSE ^1^	*p* Value	
	Winter	Spring	Summer	A_org	B_org	C	D	E	F		S	Fa
Dry Matter (DM), %	61.0	63.0	63.8	60.5	63.2	61.4	62.4	63.0	65.3	2.004	0.0915	0.1541
Protein, % DM	46.0	46.6	45.1	47.7	44.4	46.7	45.7	46.6	44.6	1.501	0.2585	0.1342
Fat, % DM	40.4	42.6	42.0	38.6	41.9	42.6	42.2	41.0	43.7	2.515	0.3271	0.2891
Ash, % DM	7.98	7.62	8.47	6.43 ^b^	9.40 ^a^	7.13 ^b^	7.13 ^b^	8.93 ^a^	9.10 ^a^	0.757	0.1991	0.0024
NaCl, % DM	3.65	3.44	3.98	2.31 ^b^	5.15 ^a^	2.70 ^b^	2.56 ^b^	4.76 ^a^	5.64 ^a^	0.622	0.3512	0.0004

The results indicate the mean values of three measurements performed on each cheese sample. ^1^ RMSE, root mean standard error. On rows: ^a^ and ^b^ = *p* ≤ 0.05.

**Table 4 animals-12-03476-t004:** Effects of production season and farm on fatty acids profile (g/100 g FA) of Caciocavallo Palermitano cheeses.

	Season (S)			Farm (Fa)						RMSE ^1^	*p* Value	
	Winter	Spring	Summer	A_org	B_org	C	D	E	F		S	Fa
C4:0	3.20	3.36	3.09	3.19	3.27	3.16	3.01	3.58	3.08	0.314	0.3654	0.3741
C6:0	2.28 ^a^	2.33 ^a^	1.91 ^b^	2.14	2.19	2.01	2.16	2.35	2.19	0.226	0.0193	0.6615
C7:0	0.08 ^c^	0.29 ^a^	0.15 ^b^	0.15	0.19	0.21	0.14	0.18	0.19	0.041	<0.0001	0.3382
C8:0	1.39 ^a^	1.46 ^a^	1.08 ^b^	1.23	1.33	1.18	1.32	1.36	1.43	0.165	0.0062	0.5483
C9:0	0.09 ^b^	0.20 ^a^	0.14 ^b^	0.12	0.16	0.17	0.10	0.14	0.16	0.042	0.0023	0.3941
C10:0	2.80 ^a^	3.07 ^a^	2.15 ^b^	2.46	2.73	2.41	2.71	2.67	3.03	0.420	0.0096	0.5553
C11:0	0.37	0.38	0.31	0.31	0.38	0.33	0.36	0.36	0.38	0.054	0.1134	0.5311
C12:0	3.05 ^a^	3.37 ^a^	2.37 ^b^	2.67	3.08	2.72	2.97	2.94	3.18	0.441	0.0081	0.6879
C12:1	0.08	0.09	0.07	0.07	0.09	0.08	0.08	0.08	0.09	0.015	0.1656	0.6450
C13:0	0.21 ^b^	0.30 ^a^	0.26 ^a^	0.24	0.25	0.28	0.26	0.23	0.26	0.031	0.0018	0.3742
C14:0	11.2	11.6	10.5	11.0	11.3	10.9	10.8	11.3	11.3	0.938	0.2009	0.9582
C14:1	0.83	0.85	0.82	0.76	0.90	0.75	0.89	0.84	0.84	0.112	0.8819	0.4939
C15:0	1.69 ^b^	1.76 ^b^	1.96 ^a^	1.99 ^a^	1.72 ^b^	2.10 ^a^	1.61 ^b^	1.73 ^b^	1.65 ^b^	0.124	0.0093	0.0035
C15:1	0.07	0.08	0.07	0.09	0.07	0.07	0.07	0.07	0.08	0.013	0.1281	0.3220
C16:0	26.7 ^a^	24.3 ^b^	28.2 ^a^	25.3	26.6	26.3	26.4	27.2	26.8	1.631	0.0061	0.7842
C16:1	1.03	1.54	1.39	1.12	1.48	1.53	1.10	1.28	1.40	0.392	0.1175	0.6803
C17:0	1.28	0.96	1.17	1.46	1.02	1.19	1.34	0.98	0.82	0.361	0.3326	0.3261
C17:1	0.24	0.25	0.32	0.28	0.25	0.33	0.29	0.25	0.22	0.048	0.0687	0.1798
C18:0	11.5	10.5	11.5	11.7	10.6	10.9	10.9	11.5	11.4	1.208	0.2652	0.8461
C18:1c9 OA	20.4 ^a^	17.2 ^b^	21.8 ^a^	20.3	19.4	19.1	20.6	18.8	19.6	1.720	0.0024	0.4312
C18:1t11 TVA	2.95 ^b^	4.95 **^a^**	2.23 ^b^	3.74	3.78	3.93	2.60	3.36	2.86	0.652	<0.0001	0.1537
Other C18:1	2.72 ^b^	3.54 ^a^	2.94 ^b^	2.78	3.18	3.28	3.04	2.95	3.15	0.457	0.0290	0.7878
C18:2 n6 LA	1.85	1.76	1.7	1.75 ^abc^	1.36 ^c^	1.49 ^bc^	2.14 ^a^	1.52 ^bc^	1.93 ^ab^	0.251	0.0753	0.0252
Other C18:2	1.03	1.30	0.71	1.17	1.04	1.48	1.03	0.98	1.00	0.397	0.0598	0.6517
C18:2c9t11 RA CLA	1.10 ^b^	1.92 ^a^	0.87 ^b^	1.38	1.52	1.56	1.03	1.25	1.04	0.268	0.0001	0.1324
Other C18:2 CLnA	0.08 ^b^	0.45 ^a^	0.13 ^b^	0.30	0.19	0.28	0.16	0.26	0.12	0.0948	<0.0001	0.2164
C18:3 n3 ALA	0.78 ^b^	1.11 ^a^	0.55 ^b^	1.10	0.66	0.97	0.70	0.76	0.70	0.221	0.0048	0.1781
C18:3 n6 GLA	0.15 ^b^	0.12 ^b^	0.24 ^a^	0.17	0.20	0.17	0.17	0.17	0.13	0.027	<0.0001	0.1923
C20:0	0.22	0.18	0.23	0.25	0.21	0.24	0.22	0.25	0.23	0.038	0.0602	0.7129
C20:1c11	0.02	0.02	0.04	0.02	0.03	0.02	0.03	0.04	0.02	0.022	0.0553	0.6895
C20:2 n6	0.04 ^b^	0.13 ^a^	0.01 ^b^	0.09	0.09	0.08	0.03	0.04	0.02	0.047	0.0024	0.3427
C20:3 n6	0.08	0.06	0.07	0.08	0.07	0.07	0.09	0.05	0.07	0.016	0.1671	0.1862
C20:4 n6	0.13	0.10	0.14	0.13	0.12	0.12	0.14	0.10	0.14	0.030	0.1915	0.6144
C20:5 n3 EPA	0.11	0.07	0.10	0.10	0.11	0.11	0.05	0.11	0.09	0.040	0.3447	0.5128
C22:0	0.10 ^b^	0.10 ^b^	0.16 ^a^	0.12	0.12	0.13	0.10	0.14	0.11	0.022	0.0004	0.3300
C22:5 n3 DPA	0.00 ^b^	0.08 ^a^	0.01 ^b^	0.02	0.06	0.02	0.07	0.02	0.00	0.056	0.0449	0.6331
C22:6 n3 DHA	0.08	0.17	0.13	0.15	0.11	0.14	0.10	0.06	0.17	0.093	0.2744	0.7124
C24:0	0.01 ^b^	0.10 ^a^	0.10 ^a^	0.10	0.08	0.06	0.05	0.07	0.04	0.022	<0.0001	0.1246
SFA	66.2	64.2	65.5	64.4	65.3	64.4	64.5	66.9	66.2	2.931	0.4951	0.8236
MUFA	28.3	28.5	29.7	29.1	29.2	29.1	29.8	27.7	28.3	2.167	0.5030	0.8648
PUFA	5.44 ^b^	7.29 ^a^	4.77 ^b^	6.45	5.52	6.49	5.74	5.35	5.43	0.866	0.0014	0.4395
n3 PUFA	0.98 ^b^	1.44 ^a^	0.79 ^b^	1.38	0.93	1.25	0.93	0.97	0.97	0.264	0.0050	0.2448
n6 PUFA	2.25 ^a^	1.88 ^b^	2.25 ^a^	2.22 ^ab^	1.84 ^b^	1.93 ^b^	2.58 ^a^	1.89 ^b^	2.30 ^a^	0.254	0.0461	0.0309
n6/n3	2.55 ^a^	1.32 ^b^	3.04 ^a^	1.77 ^b^	2.14 ^ab^	1.62 ^b^	3.44 ^a^	2.16 ^ab^	2.68 ^ab^	0.756	0.0078	0.0479
TI	2.70	2.29	2.69	2.38	2.58	2.44	2.52	2.79	2.66	0.336	0.1065	0.7073
HPI	0.43	0.43	0.45	0.46	0.43	0.45	0.46	0.41	0.42	0.065	0.7718	0.8710

The results indicate mean values of three measurements performed on each cheese. ^1^ pRMSE, root mean standard error; OA = oleic acid; TVA = trans vaccenic acid; LA = linoleic acid; CLnA = conjugated linolenic acid isomers; RA = rumenic acid; CLA = conjugated linoleic acid; ALA = α-linolenic acid; GLA = γ-linolenic acid; EPA = eicosapentaenoic acid; DPA = docosapentaenoic acid; DHA = docosahexaenoic acid. SFA = saturated fatty acids; MUFA = monounsaturated fatty acids; PUFA = polyunsaturated fatty acids; HPI = health promoting index; TI = thrombogenic index. On rows: ^a^, ^b^ and ^c^ = *p* ≤ 0.05.

**Table 5 animals-12-03476-t005:** Influences of the production season and farm on ∑PAH*, ∑PCB*, ∑ndl-PCB* and PCB_TEQ* (ng/g fat). The * symbols refer to the values normalized per gram lipid amount.

Variable	Season **			Farm						RMSE ^1^	*p* Value	
	W	S	Su	A_org	B_org	C	D	E	F		S	F
∑PAH*	57.07 ^A^	36.25 ^AB^	10.22 ^B^	20.26	21.82	61.30	36.20	33.80	33.69	26.68	0.0375	0.5086
∑PCB*	36.19 ^a^	4.68 ^b^	3.73 ^b^	5.29	5.097	11.84	7.37	24.95	34.65	23.76	0.0653	0.5784
∑ndl-PCB*	31.67 ^a^	2.79 ^b^	1.67 ^b^	1.64	2.00	9.86	6.21	20.61	31.94	20.94	0.0543	0.4749
PCB_TEQ*	0.001	0.002	0.001	0.002	0.002	0.002	0.001	0.002	0.002	0.000	0.4519	0.8928

** W = Winter; S = Spring; Su = Summer; ^1^ RMSE, root mean standard error; on rows: ^A^ and ^B^ = *p* ≤ 0.05, ^a^ and ^b^, = *p* ≤ 0.10. PAH: polycyclic aromatic hydrocarbons; PCB: polychlorinated biphenyls; ndl-PCB: not dioxin like polychlorinated biphenyls; PCB_TEQ: dioxin-like polychlorinated biphenyl was referred to tetrachloro-dibenzo-p-dioxin (TCDD).

**Table 6 animals-12-03476-t006:** Cancer risk, hazard quotient, and hazard index due to contaminated cheese ingestion. In bold values greater than 1 × 10^−6^ and 1.

Sample ID	Farm	Season	Cancer Risk			Hazard Quotient			HI
			B(a)P_TEQ	PCB_TEQ	ΣndlPCB	B(a)P_TEQ	PCB_TEQ	ΣndlPCB	HI = Σ(HQ)
Cheese_1	A_org	Winter	1.06 × 10^−8^	**1.88 × 10^−5^**	3.10 × 10^−7^	5.36 × 10^−4^	4.17 × 10^−1^	1.81 × 10^−2^	4.36 × 10^−1^
Cheese_2	B_org	Winter	6.95 × 10^−9^	**2.01 × 10^−5^**	3.53 × 10^−7^	3.52 × 10^−4^	4.46 × 10^−1^	2.06 × 10^−2^	4.67 × 10^−1^
Cheese_3	C	Winter	3.03 × 10^−8^	**2.66 × 10^−5^**	**3.96 × 10^−6^**	1.54 × 10^−3^	5.91 × 10^−1^	2.31 × 10^−1^	8.24 × 10^−1^
Cheese_4	D	Winter	2.88 × 10^−8^	**2.27 × 10^−5^**	**3.39 × 10^−6^**	1.46 × 10^−3^	5.05 × 10^−1^	1.98 × 10^−1^	7.05 × 10^−1^
Cheese_5	E	Winter	2.07 × 10^−8^	**3.84 × 10^−5^**	**1.25 × 10^−5^**	1.05 × 10^−3^	8.54 × 10^−1^	7.29 × 10^−1^	**1.58 × 10^0^**
Cheese_6	F	Winter	2.30 × 10^−8^	**2.85 × 10^−5^**	**1.76 × 10^−5^**	1.16 × 10^−3^	6.33 × 10^−1^	**1.03 × 10^0^**	**1.66 × 10^0^**
Cheese_7	A_org	Spring	1.06 × 10^−8^	**2.50 × 10^−5^**	2.36 × 10^−7^	5.39 × 10^−4^	5.55 × 10^−1^	1.38 × 10^−2^	5.70 × 10^−1^
Cheese_8	B_org	Spring	1.60 × 10^−8^	**2.88 × 10^−5^**	1.13 × 10^−7^	8.13 × 10^−4^	6.41 × 10^−1^	6.60 × 10^−3^	6.48 × 10^−1^
Cheese_9	C	Spring	1.90 × 10^−8^	**1.88 × 10^−5^**	**1.31 × 10^−6^**	9.62 × 10^−4^	4.18 × 10^−1^	7.67 × 10^−2^	4.96 × 10^−1^
Cheese_10	D	Spring	8.35 × 10^−9^	**1.94 × 10^−5^**	1.88 × 10^−7^	4.24 × 10^−4^	4.30 × 10^−1^	1.10 × 10^−2^	4.42 × 10^−1^
Cheese_11	E	Spring	1.49 × 10^−8^	**2.02 × 10^−5^**	4.55 × 10^−7^	7.54 × 10^−4^	4.49 × 10^−1^	2.66 × 10^−2^	4.76 × 10^−1^
Cheese_12	F	Spring	1.54 × 10^−8^	**2.30 × 10^−5^**	7.21 × 10^−7^	7.83 × 10^−4^	5.12 × 10^−1^	4.20 × 10^−2^	5.54 × 10^−1^
Cheese_13	A_org	Summer	1.09 × 10^−8^	**2.93 × 10^−5^**	4.42 × 10^−7^	5.55 × 10^−4^	6.52 × 10^−1^	2.58 × 10^−2^	6.79 × 10^−1^
Cheese_14	B_org	Summer	1.17 × 10^−8^	**2.46 × 10^−5^**	8.22 × 10^−7^	5.94 × 10^−4^	5.46 × 10^−1^	4.79 × 10^−2^	5.95 × 10^−1^
Cheese_15	C	Summer	8.74 × 10^−9^	**2.07 × 10^−5^**	2.49 × 10^−7^	4.44 × 10^−4^	4.61 × 10^−1^	1.46 × 10^−2^	4.76 × 10^−1^
Cheese_16	D	Summer	1.08 × 10^−8^	**2.16 × 10^−5^**	2.75 × 10^−7^	5.47 × 10^−4^	4.80 × 10^−1^	1.61 × 10^−2^	4.97 × 10^−1^
Cheese_17	E	Summer	8.74 × 10^−9^	**2.11 × 10^−5^**	2.56 × 10^−7^	4.44 × 10^−4^	4.70 × 10^−1^	1.50 × 10^−2^	4.85 × 10^−1^
Cheese_18	F	Summer	7.02 × 10^−9^	**2.09 × 10^−5^**	1.16 × 10^−7^	3.56 × 10^−4^	4.65 × 10^−1^	6.79 × 10^−3^	4.72 × 10^−1^
min			6.95 × 10^−9^	**1.88 × 10^−5^**	1.13 × 10^−7^	3.52 × 10^−4^	4.17 × 10^−1^	6.60 × 10^−3^	4.36 × 10^−1^
max			3.03 × 10^−8^	**3.84 × 10^−5^**	1.76 × 10^−5^	1.54 × 10^−3^	8.54 × 10^−1^	**1.03 × 10^0^**	**1.66 × 10^0^**
Average			1.46 × 10^−8^	**2.38 × 10^−5^**	**2.41 × 10^−6^**	7.40 × 10^−4^	5.29 × 10^−1^	1.40 × 10^−1^	6.70 × 10^−1^
Standard deviation			7.16 × 10^−9^	**5.03 × 10^−6^**	**4.81 × 10^−6^**	3.63 × 10^−4^	1.12 × 10^−1^	2.81 × 10^−1^	3.62 × 10^−1^

## Data Availability

All data included in this study are available upon request by contacting the corresponding author.

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
