# Peer review of "Persistent Organic Pollutants and Fatty Acid Profile in a Typical Cheese from Extensive Farms: First Assessment of Human Exposure by Dietary Intake"

_animals, 2022, doi:10.3390/ani12243476_

Round 1

Reviewer 1 Report

The manuscript is interesting scientific contributions to the knowledge of the persistent organic pollutants and fatty acid profile in a typical cheese from extensive farms: first assessment of human exposure by dietary intake. The consumption of pasture can improve the quality of animal products, improving the fatty acids profile and transferring nutraceutical compounds in milk and cheese with consequential beneficial effects on human health. In this regard, the aim this study was to evaluate the risk and benefits related to the consumption of typical stretched cheeses, considering their fatty acid profile and the persistent organic pollutants content. The paper has high scientific level, the experiment is well designed, the discussion is consistent and the final conclusions are interesting. Therefore, the manuscript may be published in Animals making minor revision:

Suggestions for edition as well as some comments are the following:

Abstract

Please rewrite the abstract giving more information about the obtained results

 Lines 285-286 “In fact, NaCl expressed as g/ 100 g cheese, resulted higher in cheeses with a higher DM level, as also observed in previous studies”. Please, check this affirmation according to the data from table 3 (season effect since samples from spring had lower NaCl compared with samples from winter).

 Conclusion

Please summarize the conclusions. They are very long.

 I hope that my comments can improve the manuscript.

Author Response

The authors are really grate for suggestions and comments to improve the manuscript quality.
In attach the responses.
Sincerly the authors.

Reviewer 2 Report

This is a very interesting, relevant study and potentially very useful - but it does have limitation,  both in what was done and how it is explained.  

The attached shows my concerns and makes suggestions how it might be improved.  There are sweeping statements about composition not always with a supporting reference and a few examples of vague terms like 'etc'.  

The language needs to be improved to make it easier to read.  It is also very long and I wonder (since it covers 2 very different topics) if it might be better presented as 2 related papers, rather than try to cover everything together - making discussion challenging.

I hope it can be transformed into a (possibly 2) readable, accessible paper(s) 

Author Response

The authors are really grate for suggestions and comments to improve the manuscript quality.
In attach the responses.
Sincerly the authors

Reviewer 3 Report

animals-2008064

Title:  Persistent organic pollutants and fatty acid profile in a typical cheese from extensive farms: first assessment of human exposure by dietary intake

The objective: This experiment aims to evaluate the risk and benefits related to the consumption of typical stretched cheeses, considering their fatty acid (FA) profile and the persistent organic pollutants (POPs) content.

Abstract:

L38: Specify a research trial plan, please.

L59-62: How is the above sentence related to this sentence? should be written to be linked.

L66-68: Modify sentence; Therefore, the foodstuff can present complex mixtures of these pollutant compounds, related to environmental quality, and animal exposure, causing bioaccumulation consequently in humans due to metabolism capacities [1-27].

L103-115: What is the research gap of this experiment?

L110-115: The authors should provide the hypothesis of this research.

Materials and Methods

L119: Specify the species and breed of the experimental animal.

L120-121: The authors write a detailed description of the experimental design, divided into how many treatments, and replications there were.

L121: Modify word; Eastern Sicily

L122-127: Tell me the source, the level of concentrated/roughage feed studied, how to calculate it, and whether it is sufficient for the needs of the animal or not.

L122-127: The authors write a detailed description of the experimental difference in the chemical composition of animal feed, and how to sample feed randomly.

L150-158: The author should specify the total time to try the experiment, how many intervals, how many days each period, and what days the data was collected.

L251: Table 2: The author identifies the meanings of dl-PCB_TEQ (TCDD) and ndl-PCB (below the table), please.

Result and discussions

L276-287: The authors explained each section separately for better understanding and clarity. Initially, it should be noted that the experimental results describe the main factors. What are the treatments where there are statistically different values? and report the average value of the parameters to be complete. What treatment is high?

L280-286: Why do ash and NaCl values from the same organic farm have different values?

L289-299: The authors explained each section separately for better understanding and clarity. Initially, it should be noted that the experimental results describe the main factors. What are the treatments where there are statistically different values? and report the average value of the parameters to be complete. What treatment is high?

L291: Why report the myristic (C14:0) because the data have shown non-significant.

L294-336: The authors explain in full the data showing the statistical significance of the Table 4 values and the mechanism for understanding and clarity, and the discussion should explain each factor one by one. What treatment is high?

L551: Table 5: The author identifies the meanings of PAH, PCB, ndl-PCB, PCB_TEQ, W, S, Su, S, and F (below the table), please.

Conclusion

L568-604: It should be summarized briefly to make it easy to understand, and the suggestion or conclusion should be used.

Author Response

(The authors gave the same response as above.)

Round 2

Reviewer 3 Report

The writers revised it in response to all of my comments, and hence I recommend that it be accepted in its current version.